# The asymmetry of female meiosis reduces the frequency of inheritance of unpaired chromosomes

Daniel B Cortes[1], Karen L McNally[1], Paul E Mains[2], Francis J McNally[1]*

[1]Department of Molecular and Cellular Biology, University of California, Davis, Davis, United States; [2]Department of Biochemistry and Molecular Biology, University of Calgary, Calgary, Canada

**Abstract** Trisomy, the presence of a third copy of one chromosome, is deleterious and results in inviable or defective progeny if passed through the germ line. Random segregation of an extra chromosome is predicted to result in a high frequency of trisomic offspring from a trisomic parent. *Caenorhabditis elegans* with trisomy of the X chromosome, however, have far fewer trisomic offspring than expected. We found that the extra X chromosome was preferentially eliminated during anaphase I of female meiosis. We utilized a mutant with a specific defect in pairing of the X chromosome as a model to investigate the apparent bias against univalent inheritance. First, univalents lagged during anaphase I and their movement was biased toward the cortex and future polar body. Second, late-lagging univalents were frequently captured by the ingressing polar body contractile ring. The asymmetry of female meiosis can thus partially correct pre-existing trisomy.

## Introduction

*For correspondence: fjmcnally@ ucdavis.edu

**Competing interests:** The authors declare that no competing interests exist.

**Reviewing editor**: Anthony A Hyman, Max Planck Institute of Molecular Cell Biology and Genetics, Germany

During female meiosis, a G2 oocyte containing four genome copies undergoes two asymmetric cell divisions depositing one genome in a single haploid egg, while the other three genomes are segregated into polar bodies. These divisions are mediated by meiotic spindles that are asymmetrically positioned against the oocyte cortex with the pole-to-pole axis of the spindle perpendicular to the cortex. Both the inheritance of only one of the four genome copies and the distinct perpendicular positioning of the meiotic spindle are remarkably conserved among animal phyla suggesting a selective advantage (*Maro and Verlhac, 2002*; *Fabritius et al., 2011a*; *Maddox et al., 2012*).

Several advantages of asymmetric meiosis have been suggested previously, yet none are applicable to all animals. Asymmetric meiotic spindle positioning maximizes the volume of a single egg, helps prevent interference with the meiotic spindle by the sperm aster (*McNally et al., 2012*), and preserves predetermined embryonic polarity gradients. Here, we suggest a previously un-recognized advantage of asymmetric meiosis, the ability of meiotic spindles to correct trisomy by preferentially depositing the extra chromosome copy into a polar body.

Accurate segregation of homologous chromosomes to opposite spindle poles depends on a physical attachment, or chiasma, between homologous chromosomes. A chiasma consists of a crossover, which holds the two homologous chromosomes together in a bivalent so that kinetochores can be properly oriented toward opposite poles (*Moore and Orr-Weaver, 1998*; *Miller et al., 2013*). When a chiasma does not form, univalent chromosomes may maintain sister cohesion and move to poles independent of their homologs at anaphase I as can occur in *Saccharomyces cerevisiae* (*Buonomo et al., 2000*). If a univalent chromosome biorients, loses cohesion, and segregates sister chromatids at anaphase I (e.g., *Nicklas and Jones, 1977*; *Lemaire-Adkins and*

**eLife digest** Inside cells, DNA is found packaged into structures called chromosomes. Most human and animal cells contain two sets of chromosomes, one inherited from each parent. Chromosomes from one set pair up with the equivalent chromosome from the other set. However, egg and sperm cells only contain one copy of each chromosome, so that when the egg is fertilized, the resulting cell again has two sets of chromosomes. If there are either more or fewer than two copies of a chromosome in the fertilized cell, this can cause birth defects and conditions such as Down syndrome.

An egg cell develops from a cell called an oocyte via a process called meiosis. The oocyte first duplicates its DNA so that it contains four copies of each chromosome. The oocyte then divides, and the resulting cells divide again, to produce four cells that each contains one copy of each chromosome. Only one of these cells is an egg cell: the other three are called polar bodies, and these normally self-destruct.

The tiny roundworm *C. elegans* is a model organism used to study meiosis. Worms can be hermaphrodites or males; the hermaphrodites normally have a pair of 'X' sex chromosomes. However, sometimes problems with meiosis can produce hermaphrodite worms with three X chromosomes in each of their cells. In these cells, two of the X chromosomes pair with each other as normal, and one X chromosome remains unpaired.

Cortes et al. examined meiosis in mutant worms that had an extra copy of the X chromosome by marking all the chromosomes with a fluorescent tag. This allowed the movement of the chromosomes to be tracked through images taken using a microscope. This revealed that an unpaired X chromosome moves more slowly than a normal paired set. Furthermore, the unpaired chromosomes tend to move toward the region of the oocyte that will develop into a polar body. Thus, when the oocyte divides, the unpaired chromosomes are placed in the polar body and eliminated. This mechanism improves the chance that the correct number of chromosomes will end up in the egg cell.

Women with three X chromosomes are often fertile and in most cases produce normal offspring. Further work is needed to see whether human oocytes remove extra chromosomes by a mechanism similar to that seen in the roundworms.

*Hunt, 2000*; *Kouznetsova et al., 2007*), the resulting single chromatid will segregate randomly at anaphase II. Random segregation of homologs at anaphase I or single chromatids at anaphase II should result in equal frequencies of haplo and diplo ova in the case of a trisomy (*Figure 1A*) and equal frequencies of nullo and diplo ova in the case of a crossover failure.

Deviations from random segregation are suggested by observations of X chromosomes in *Caenorhabditis elegans*. In *C. elegans*, the single unpaired X chromosome from an XXX mother is inherited with unexpectedly low frequency with twice as many haploX ova produced as diploX ova (*Hodgkin et al., 1979*). HIM-8 is a zinc finger protein that binds to specific DNA sequences that are enriched on the X chromosome. *him-8* mutants have a pairing defect that is completely specific for the X chromosome, resulting in two X univalents and five autosomal bivalents in 95% of diakinesis oocytes (*Phillips et al., 2005*). If the two X univalents segregated randomly, *him-8* mutants would be expected to produce equal frequencies of nulloX ova and diploX ova. However, *Hodgkin et al. (1979)* demonstrated a fivefold preponderance of nulloX ova over diploX ova in *him-8*. Using sex-reversed *him-8* XX males, these authors showed the opposite effect in spermatogenesis. Rather than producing the 50% haploX, 25% diploX, 25% nulloX sperm expected from random segregation, *him-8* XX males produced 86% haploX, 3% diploX, 11% nulloX sperm, indicating symmetric distribution of univalents during male meiosis. Thus, achiasmate maternal X chromosomes are inherited with unexpectedly low frequency in worms.

Five mechanisms might reduce the frequency of trisomic offspring from trisomic or *him-8* mothers. First, trisomic embryos might die during embryonic development resulting in under-counting of XXX offspring. This is unlikely in *C. elegans* because both XXX and *him-8* mutant mothers produce a very low frequency of dead embryos (*Hodgkin et al., 1979*; *Supplementary file 1*). A second possibility is that mitotic non-disjunction in the XXX mother results in a mosaic

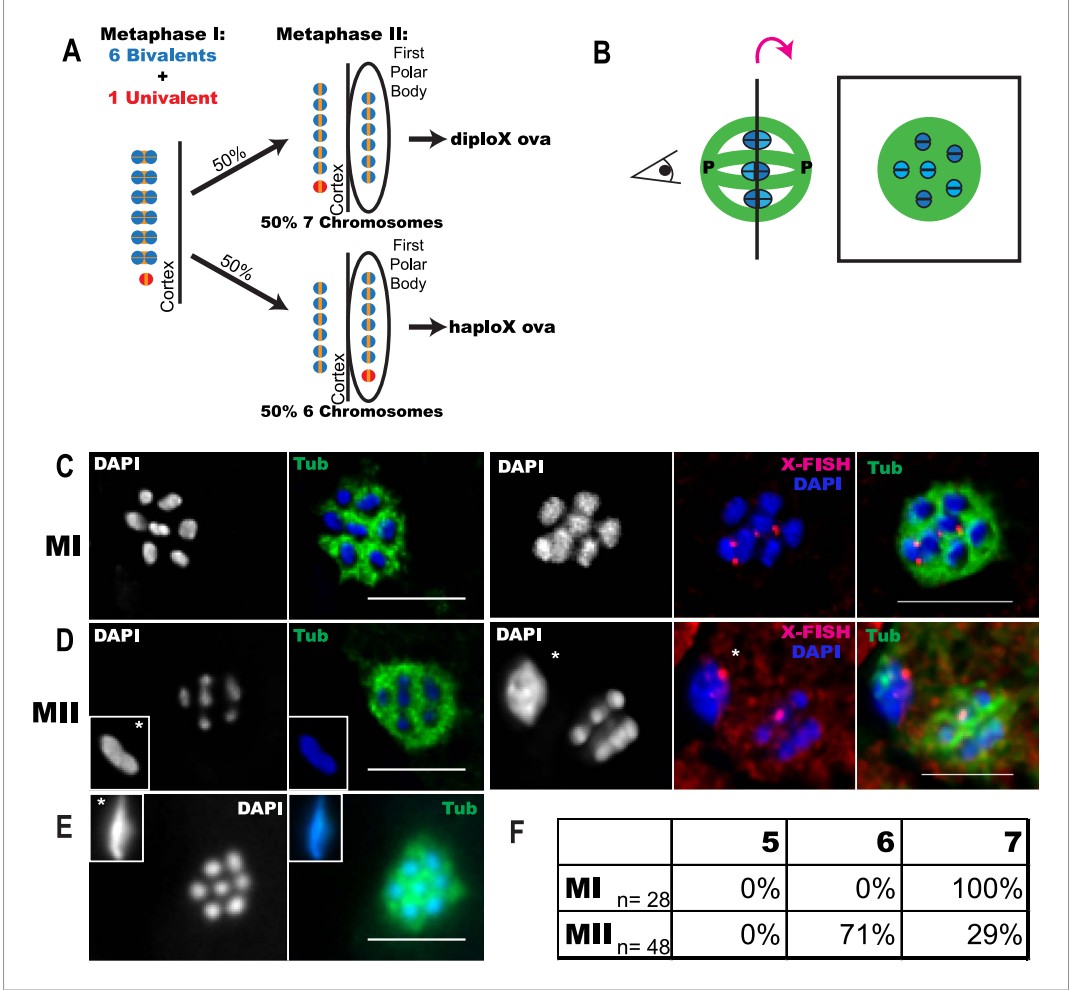

**Figure 1**. Trisomy correction during meiosis I. (**A**) Illustration showing expected outcomes of female meiosis in XXX wild-type worms, assuming the extra univalent X (red) does not lose cohesion (yellow) between sister chromatids during anaphase I and assuming random segregation. (**B**) Illustration of a spindle with chromosomes at the metaphase plate with poles marked 'P' (left) and a projection of the cross-sectional view down the pole-to-pole axis at the metaphase plate (right). (**C**–**E**) Z projections of fixed meiotic embryos viewed down the pole-to-pole spindle axis. Meiotic embryos from XXX wild-type mothers were stained with DAPI and anti-tubulin antibody. (**C**) Metaphase I spindles with 7 chromosomes; right images of X-fluorescence in situ hybridization (FISH) show two X chromosomes on the spindle. See also *Supplementary file 2*. (**D**) Metaphase II spindles with 6 chromosomes; right images of X-FISH show one X on the spindle and 2–3 foci in the polar body. See also *Supplementary file 3*. (**E**) Metaphase II spindle with 7 chromosomes. (**F**) Frequency of each spindle class among the progeny of XXX wild-type mothers. Insets show polar bodies, marked by asterisks, which were used to identify metaphase II spindles. Bar = 5 μm.

gonad that contains both diploX and triploX oocytes. Selective apoptosis of XXX germ line cells (*Bhalla and Dernburg, 2005*) would then enrich for XX germ line cells. This does not contribute to the segregation bias in *C. elegans*, as the most mature diakinesis oocytes in *him-8* and wild-type XXX worms have 7 rather than 6 DAPI-staining bodies (*Phillips et al., 2005*; this study). A third possibility is that a univalent present during metaphase I or a single chromatid present during metaphase II would be broken or otherwise degraded during anaphase. A fourth possibility is that many XXX progeny look normal because of the stochastic nature of dosage compensation and thus are undercounted. A fifth possibility is that univalent chromosomes present at metaphase I are preferentially placed in the first polar body. Here, we demonstrate that indeed biased deposition of univalent X chromosomes into the first polar body reduces the frequency of trisomic zygotes resulting from oocytes with unpaired X chromosomes.

## Results

### XXX wild-type oocytes preferentially lose the achiasmate X chromosome between metaphase I and metaphase II

Elimination of the extra chromosome from an oocyte starting with a trisomy would result in rescue to a euploid state. It has previously been shown that *C. elegans* XXX wild-type oocytes have a paired bivalent X and an unpaired univalent X chromosome in pachytene (*Goldstein, 1984*). We picked wild-type XXX adult hermaphrodites from the progeny of an XXX strain (AV494, *Mlynarczyk-Evans et al., 2013*) based on their characteristic dumpy morphology as described by *Hodgkin et al. (1979)*. Meiotic embryos from XXX mothers were fixed and stained for microtubules and DNA. Chromosomes are well separated by bundles of microtubules during *C. elegans* female meiotic metaphase. This unique morphology facilitates counting of individual chromosomes on the metaphase plate when viewed down the pole-to-pole axis of the spindle (*Figure 1B*). We found that 100% of metaphase I meiotic embryos from XXX wild-type worms had 7 DAPI-staining bodies on the spindle (*Figure 1C,F*), consistent with 6 bivalents and a single univalent X. Two chromosomes were labeled with an X-specific fluorescence in situ hybridization (FISH) probe in these spindles (*Figure 1C*, *Supplementary file 2*). This result shows that a mosaic gonad resulting from mitotic nondisjunction cannot explain the low frequency of XXX offspring from XXX worms. If the univalent segregated randomly during anaphase I, 50% of metaphase II spindles should have 6 DAPI-staining bodies (6 bivalents) and 50% should have 7 DAPI-staining bodies (6 bivalents and 1 univalent). Instead, 71% of metaphase II spindles contained only 6 DAPI-staining bodies and only 29% contained 7 DAPI-staining bodies (*Figure 1D–F*). These frequencies match the 2:1 ratio of X to XX ova previously interpreted from genetic studies (*Hodgkin et al., 1979*) and are significantly different than the 50% expected from random segregation (one-tailed $p = 0.004$, Pearson's chi-squared test). This result eliminates the possibilities that XXX mothers have many XXX offspring that are undercounted due to incomplete penetrance of the XXX dumpy phenotype or that hermaphrodite nulloX sperm contributes significantly to the low frequency of XXX self-progeny. The finding that all assayed metaphase I spindles had 7 chromosomes also indicates that our method of identifying XXX worms as dumpy individuals is accurate and the high frequency of metaphase II spindles with 6 chromosomes is not a result of misidentifying diploid worms as XXX worms. FISH with an X-specific probe revealed that in 6/6 metaphase II embryos with only 6 DAPI-staining bodies, a single hybridization signal was present in the spindle and 2–3 hybridization signals were present in the polar body (*Figure 1D*; *Supplementary file 3*). Because only a single X-hybridization signal was observed in the first polar body in 5/5 spindles from diploids, these results demonstrate that single X univalents are deposited in the first polar body with greater than 50% frequency.

### X and V univalents are frequently deposited in the first polar body

To further investigate the mechanism leading to preferential loss of univalents during meiosis I, we utilized *him-8* worms as a more tractable model. It has previously been shown that diakinesis stage *him-8* oocytes have 5 autosomal bivalents and two X univalents (*Phillips et al., 2005*). If segregation of the two X univalents was random, these worms should produce equal numbers of nulloX and diploX ova. Instead, *him-8* mutants produce a fivefold higher frequency of nulloX ova over diploX ova (*Hodgkin et al., 1979*), indicating that both maternal X univalents are lost at some time after diakinesis in a large fraction of embryos. To determine when maternal X univalents are preferentially lost, we imaged both live embryos within *him-8* worms expressing GFP::tubulin and mCherry::histone (*Figure 2A–E*) and also fixed embryos stained with DAPI and anti-tubulin antibodies (*Figure 2F–J*). We assayed the number of chromosomes (defined here as DAPI-staining or mCherry:histone-positive bodies that would include univalents and bivalents) present at metaphase of meiosis I and II (*Figure 2K*). At meiosis I metaphase, 7 chromosomes were present in 96% of *him-8* embryos (*Figure 2B,K*), with the remainder having 6 chromosomes. If the two univalents segregated randomly without losing cohesion, 25% of metaphase II spindles would be expected to have 5 autosomes and no X, 50% would have 5 autosomes and 1 X, and 25% would have 5 autosomes and 2 X chromosomes. Instead, 40% of *him-8* metaphase II embryos had 5 chromosomes, 55% had 6 chromosomes, and only 5% had 7 chromosomes (*Figure 2K*). These frequencies differ significantly from those expected from unbiased segregation (chi-square test, two-tailed $p < 0.0001$), closely match the ratio of nulloX to

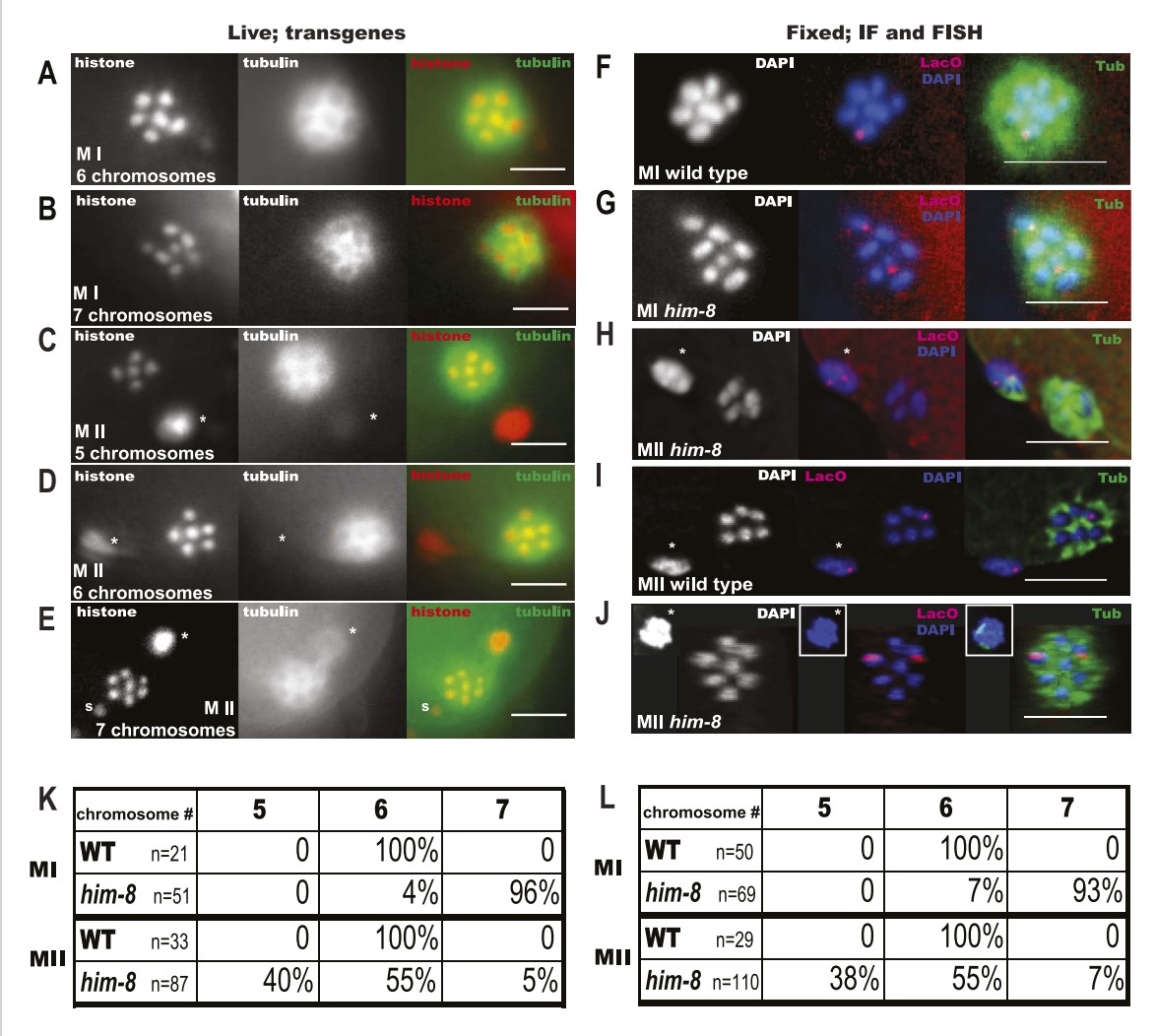

**Figure 2**. X univalents are preferentially lost between metaphase I and metaphase II in *him-8* mutants. Z projections of living (**A–E**) and fixed (**F–J**) *C. elegans* meiotic embryos viewed down the pole-to-pole spindle axis at metaphase I (**A, B, F, G**) or metaphase II (**C, D, E, H, I, J**). mCherry::Histone H2B and GFP::tubulin label the chromosomes and spindle, respectively, in live embryos. Fixed embryos were stained with DAPI, anti-tubulin antibody, and a LacO FISH probe that recognizes a LacO array integrated on the X chromosome. Asterisks indicate polar bodies. Insets show polar bodies that did not fit in the image frame. In (**E**), 's' denotes a sperm outside of the embryo. Percentages are shown for each outcome (**K, L**). Bar = 5 μm.

The following figure supplement is available for figure 2:

**Figure supplement 1**. *zim-2* embryos also deposit unpaired chromosome V univalents into the first polar body.

diploX ova inferred by *Hodgkin et al. (1979)* and support the hypothesis that the majority of X univalents are eliminated between metaphase I and metaphase II. These maternal chromosome counts are also unaffected by nulloX or diploX sperm that might contribute to phenotype-based progeny counts.

To confirm that the two chromosomes lost between these stages were indeed the X univalents, we used FISH with a lac operator probe to detect a multi-copy lac operator array integrated on the X chromosomes in a *him-8* background (*Figure 2F–J*). FISH revealed two X univalents and 7 total DAPI-staining bodies on 93% of all the *him-8* metaphase I plates (*Figure 2G,L*). At metaphase II, FISH revealed at least two X hybridization foci in the first polar body and none on the metaphase plate when the spindle had 5 DAPI-staining bodies (*Figure 2H*). When 6 DAPI-staining bodies were present on the metaphase II plate, we always observed one X hybridization focus each on the metaphase plate

and in the first polar body (*Figure 2I*). Finally, metaphase II embryos containing 7 DAPI-staining bodies had two X hybridization foci on the metaphase plate and none in the first polar body (*Figure 2J*). Together, these results demonstrate that both achiasmate X univalents are deposited into the first polar body in 40% of *him-8* embryos as compared with the 25% expected from random segregation.

To test whether achiasmate autosomes are also placed in the first polar body with higher than random frequency, we analyzed a strain with a lac operator array integrated on chromosome V and bearing a loss of function mutation in the *him-8* homolog *zim-2*, which contributes to chromosome V pairing (*Phillips and Dernburg, 2006*). Unlike the situation with *him-8* and pairing of the X, redundancy between ZIM proteins may contribute to chromosome V pairing. Phillips and Dernburg reported only 72% of diakinesis oocytes with 7 rather than 6 DAPI-staining bodies in a *zim-2* mutant, and our *zim-2* strain with lacO(V) had only 62% of diakinesis oocytes with 7 DAPI-staining bodies (*Figure 2—figure supplement 1F*). FISH revealed two distinct chromosome V hybridization foci and 7 DAPI-staining bodies on 41% of metaphase I spindles (*Figure 2—figure supplement 1A,E*). Starting with 41% achiasmate V's, random segregation should yield 10% of metaphase II embryos with both V's in the first polar body (25% of 41%). Instead, FISH revealed 27% of metaphase II embryos had five DAPI-staining bodies on the metaphase plate and chromosome V hybridization foci only in the first polar body (*Figure 2—figure supplement 1B,E*). Likewise, random segregation of achiasmate V's should yield 10% metaphase II spindles with 7 DAPI-staining bodies on the metaphase II spindle, two distinct chromosome V hybridization foci on the spindle, and none in the first polar body. Only 5% of this embryo class was observed (*Figure 2—figure supplement 1D,E*). These frequencies are significantly different than those expected from random segregation (chi-square test, two-tailed p < 0.0002). The discrepancy in the fraction of *zim-2* oocytes with 7 DAPI-staining bodies at diakinesis vs metaphase I raises the possibility that chromosomes might be systematically undercounted in *zim-2* metaphase I spindles (but not in wild-type, *him-8*, or XXX metaphase I spindles). If this is the case, the two V univalents must be positioned close together on the spindle because 0/57 *zim-2* metaphase I plates with 6 DAPI-staining bodies had two widely spaced lacO(V) FISH foci and the deviation between expected and observed nulloV metaphase II spindles would be even greater. Two results strongly indicate that the same mechanisms acting on univalent X's in *him-8* mutants also act on V univalents in the *zim-2* mutant. First, the fivefold preponderance of metaphase II spindles with 5 DAPI-staining bodies over those with 7 DAPI-staining bodies is similar to *him-8*. Second, the presence of lacO FISH signal only in the first polar body of metaphase II embryos with 5 DAPI-staining bodies on the spindle is the same in *him-8* and *zim-2*. Thus, achiasmate autosomes, like achiasmate X chromosomes, are preferentially deposited into the first polar body.

## Univalents biorient at metaphase I and tend to lag during anaphase I

To understand the mechanism by which univalent X chromosomes are preferentially deposited in polar bodies, we examined their orientation and position in the spindle. Antibodies specific for the cohesin subunit, REC-8, label a cruciform on metaphase I bivalents (*Figure 3A,B*) and a single band on metaphase II chromosomes (*Figure 3A,C*). The single REC-8 bands on wild-type metaphase II chromosomes and on *him-8* metaphase I univalents were both oriented perpendicular to the pole-to-pole axis of the spindle (*Figure 3D*), indicating that *him-8* X univalents biorient at metaphase I. *him-8* worms expressing GFP::KNL-2, which labels the *C. elegans* cup-shaped meiotic kinetochores (*Dumont et al., 2010*), were also analyzed for biorientation and yielded the same conclusion as analysis by REC-8 antibody (*Figure 3—figure supplement 1*). We also examined the localization of GFP:AIR-2, the aurora B kinase that is essential for the loss of cohesion at anaphase I and which is loaded between homologs of wild-type bivalents in a chiasma-dependent fashion (*Rogers et al., 2002*). The fluorescence intensity of GFP::AIR-2 was 2.3 times higher on bivalents than univalents at metaphase I of *him-8* embryos (*Figure 3—figure supplement 2B,C*). AIR-2 is normally re-loaded between sister chromatids at metaphase II. GFP::AIR-2 on metaphase II chromosomes was 1.7 times higher than on *him-8* metaphase I univalents (*Figure 3—figure supplement 2C*), indicating that the reduced amount of AIR-2 on metaphase I univalents was not simply a consequence of the smaller size of a univalent relative to a bivalent. AIR-2 is required for the crossover dependent, prometaphase, partial removal of REC-8 from between homologs in a wild-type bivalent, an event proposed to be essential for loss of cohesion at anaphase I (*Severson and Meyer, 2014*). Consistent with the low levels of AIR-2, *him-8* univalents had 1.7 $\pm$ 0.3 times the intensity of REC-8 staining as the

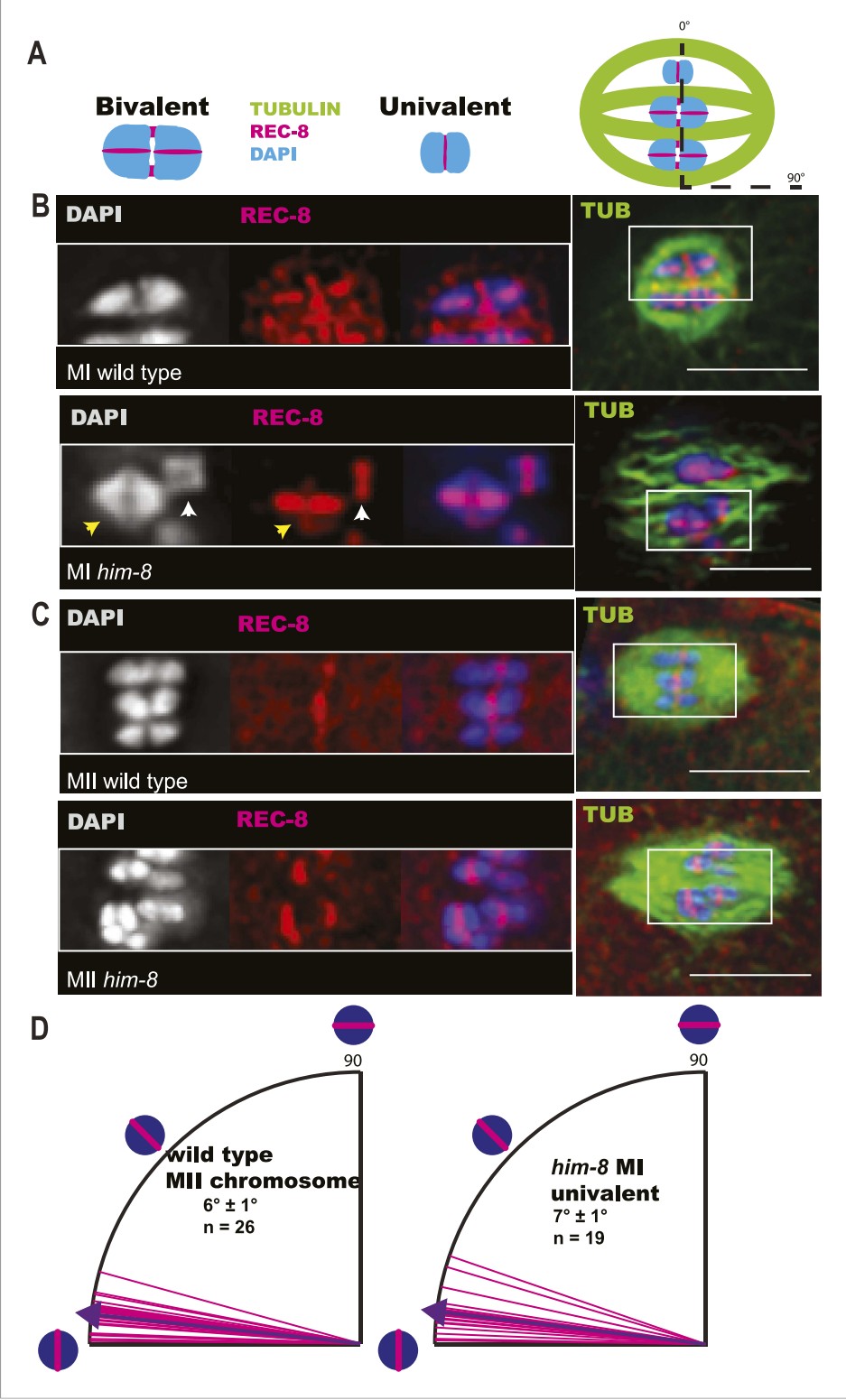

**Figure 3**. X univalents biorient at metaphase I in *him-8* embryos. (**A**) Cartoon diagram of REC-8 staining on bivalents and univalents. (**B** and **C**) Anti-REC-8 staining of metaphase I and metaphase II embryos with bivalents (yellow arrow head) and univalents (white arrow head). In *him-8* embryos, univalents at metaphase I have a single band of REC-8 with the same orientation seen on normal chromosomes at metaphase II. (**D**) Quantification of the orientation of

*Figure 3. continued on next page*

*Figure 3. Continued*

univalents by offset angle from the metaphase plate, 0° corresponds to perfect biorientation and 90° corresponds to perfect mono-orientation. Cortical pole is on the left in all images. Bar = 5 µm.

The following figure supplements are available for figure 3:

**Figure supplement 1**. Imaging of GFP::KNL-2 demonstrates that *him-8* univalent chromosomes biorient at metaphase of meiosis I.

**Figure supplement 2**. Reduced levels of AIR-2 are loaded on *him-8* X univalents at meiosis I.

inter-homolog region of bivalents in the same spindle (*Figure 3B*; n = 8 embryos, two-tailed p = 0.04, chi square relative to expected 1.0).

Because X univalents biorient at metaphase I but load half as much AIR-2, which is required for loss of cohesion at anaphase I in *C. elegans* (*Kaitna et al., 2002*; *Rogers et al., 2002*), and retain twice as much REC-8, we hypothesized that bioriented univalents might be pulled toward both spindle poles and lag on the anaphase spindle as they fail to lose cohesion. To test this possibility, we did time-lapse imaging of *him-8* embryos expressing GFP::tubulin and mCherry::histone, focusing specifically on the events of anaphase I. 90% of *him-8* embryos at anaphase I had one or two lagging chromosomes (n = 119) compared to 2% of wild-type embryos (n = 52) (*Figure 4A,B*). In 51% of living *him-8* embryos with lagging chromosomes at anaphase I, two discrete lagging chromosomes could be resolved. Each lagging chromosome eventually moved as a single unit either toward the cortex or into the embryo in 98% of embryos (n = 179) (*Figures 4B, 5A*), indicating that cohesion between sister chromatids is maintained and that univalents are not broken or destroyed during anaphase. At anaphase II, only 10% of *him-8* embryos exhibited lagging chromosomes (n = 60) and 0/22 wild-type embryos had lagging chromosomes, suggesting that lagging chromosomes are caused by the presence of univalents at meiosis I.

To confirm that the lagging chromosomes are bioriented X univalents, we used LacO(X) FISH. We found that the X-specific FISH probe labeled one or two lagging chromosomes at anaphase I of *him-8* embryos, indicating that lagging chromosomes are X univalents (13/14) (*Figure 4D*). 36% of fixed *him-8* anaphase I embryos with lagging chromosomes had two distinct FISH-positive chromosomes lagging. Another 21% had a single FISH-positive lagging body, but no other FISH-positive chromosomes on the spindle indicating that the two X univalents were likely too close to resolve in these embryos. The remaining 36% had one FISH-positive lagging chromosome and one FISH-positive chromosome in one of the main chromosome masses (one embryo had a lagging chromosome that was a bivalent). These results suggest that one or both X univalents lag in up to 90% of *him-8* embryos.

Similar results were obtained for chromosome V in *zim-2* mutants, where 40% of metaphase I embryos have univalent V's (*Figure 2—figure supplement 1*). 27% (4/15 or over half of anaphase I spindles expected to have V univalents) had a lagging chromosome. 100% (4/4) of these lagging chromosomes were chromosome V as assayed by LacO(V) FISH (*Figure 4E*). These results indicate that achiasmate autosomes lag at anaphase I, just like achiasmate X chromosomes.

## The meiotic contractile ring captures lagging X univalents in the first polar body

After establishing that lagging chromosomes were univalents, we next asked if univalents that lagged were subject to biased segregation at anaphase I. To analyze this, we conducted time-lapse imaging of embryos from *him-8* worms expressing GFP::tubulin and mCherry::histone, as well as embryos from *him-8* worms expressing these along with GFP::PH (plextrin homology domain) to label the plasma membrane (*Figure 5A*). Our time-lapse analysis revealed that 65% of lagging chromosomes eventually moved toward the cortex and the forming polar body of *him-8* embryos during anaphase I (n = 181) (*Figure 5B*), indicating that preferential expulsion of lagging univalents into the first polar body could contribute to the higher than random frequency of metaphase II spindles with 5 autosomes and no X. Because the polar body contractile ring ingresses inward toward the midpoint of the late anaphase spindle where it normally scissions, we hypothesized that the preferential resolution of lagging chromosomes toward the cortex might result from inhibition of contractile ring scission

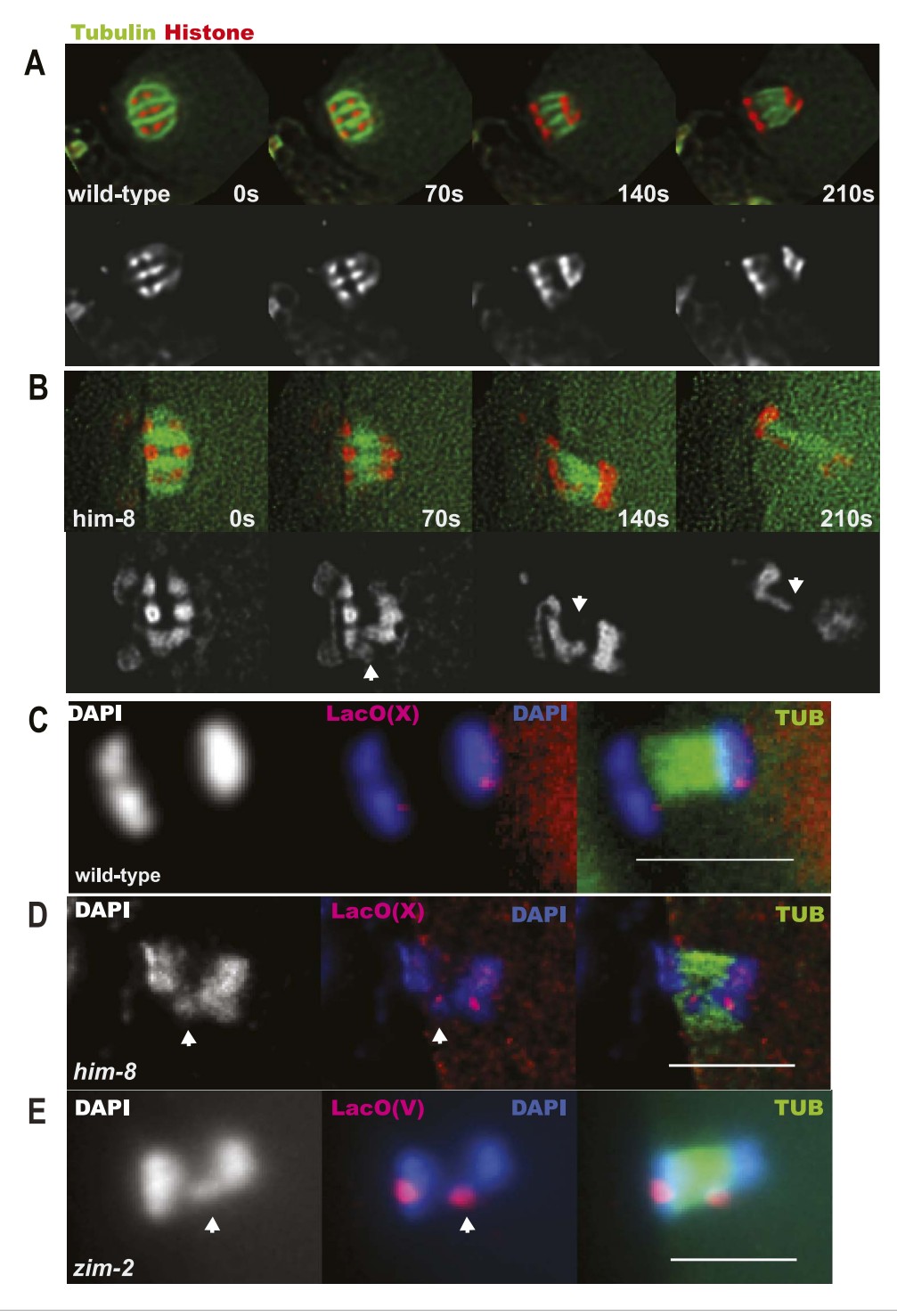

**Figure 4**. X univalents lag at anaphase I. (**A**) Time-lapse images of a living wild-type embryo undergoing anaphase I show chromosomes separating as two distinct masses. (**B**) Time-lapse images of a living *him-8* embryo show a lagging chromosome at anaphase I. (**C–E**) Z projections of fixed anaphase I embryos. (**C**) LacO FISH labeling of a wild-type strain with a LacO array integrated on the X chromosome shows normal segregation of two X homologs from one X bivalent. (**D**) LacO FISH shows that a lagging chromosome in *him-8* is the X. (**E**) LacO FISH labeling of a *zim-2* strain with a LacO array integrated on chromosome V showing a univalent V lagging at anaphase I. Cortical pole is to the left in all images. Bar = 5 µm.

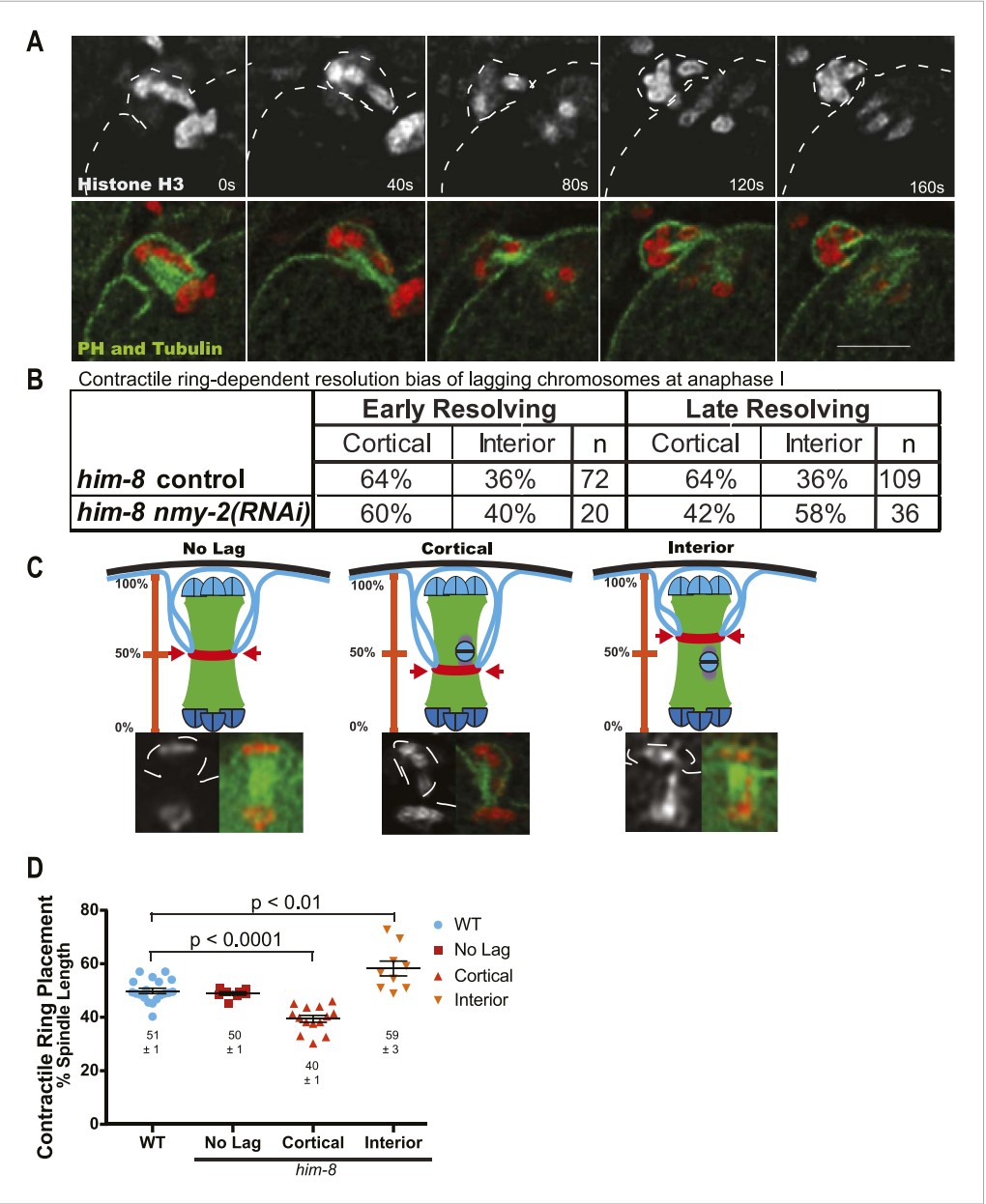

Figure 5. The contractile ring moves inward past the lagging chromosomes of *him-8* embryos. (**A**) Time-lapse sequence of anaphase I in a *him-8* strain with GFP::PH, GFP::Tubulin, and mCherry::Histone H2B. The plasma membrane ingresses past the lagging chromosomes to engulf them in the polar body. (**B**) Fraction of *him-8* anaphase I embryos in which a lagging chromosome eventually resolved toward the cortex or eventually resolved into the embryo (interior). Lagging univalents resolved more frequently toward the cortex during both early and late anaphase. Depletion of NMY-2, the myosin required for polar body formation, eliminated only the late anaphase bias. Pairwise two-tailed p values by Fisher's exact test: *him-8* late vs *him-8 nmy-2(RNAi)* late = 0.02, *him-8* early vs *him-8 nmy-2(RNAi)* early = 0.80, *him-8* early vs *him-8* late = 1.0, *him-8 nmy-2(RNAi)* early vs *him-8 nmy-2(RNAi)* late = 0.26. p values from Pearson's chi-squared test: *him-8* late vs 50% = 0.003, *him-8 nmy-2(RNAi)* late vs 50% = 0.32, *him-8* early vs 50% = 0.02, *him-8 nmy-2(RNAi)* early vs 50% = 0.38. (**C**) Top, diagram illustrating how the position of scission by the contractile ring along the pole-to-pole spindle axis was scored. Bottom, representative images from time-lapse sequences showing scission at different positions along the length of the spindle. (**D**) Average position of contractile ring scission along the pole-to-pole spindle axis in wild-type embryos and in *him-8* embryos with no lagging chromosomes, lagging chromosomes that end up at the cortex (cortical), or lagging chromosomes that end up in the embryo (interior). Bar = 5 μm.

until the ring ingresses past lagging chromosomes (*Figure 5C*). In wild type, ingression of the polar body contractile ring initiates when homologs have separated by 2.3 µm and polar body scission completes when homologs have separated by 5.6 µm (*Fabritius et al., 2011b*). The bias of univalents that moved toward the cortex before initiation of contractile ring ingression could not be caused by engulfment by the polar body. Therefore, we separated lagging univalents into two categories, early-resolving univalents that moved to one pole while the main chromosome masses were separated by less than 4.0 µm and late-resolving univalents that moved to one pole only after the main chromosome masses were separated by greater than 4.0 µm. If late-resolving univalents were engulfed during polar body formation, elimination of contractile ring activity would reduce the fraction of lagging chromosomes resolving toward the cortex. Indeed, RNAi depletion of the non-muscle myosin, NMY-2, which causes complete loss of cortical furrowing and polar body formation (*Fabritius et al., 2011b*), in *him-8* embryos resulted in a significant (p = 0.02) reduction in the percentage of late-lagging univalents resolving toward the cortex from 64% to 43% (*Figure 5B*).

As a complementary approach, we asked if more rapid polar body ring ingression would have the opposite effect of NMY-2 depletion. We previously showed that the depletion of the myosin phosphatase, MEL-11, doubled the rate of polar body ring ingression (*Fabritius et al., 2011b*). Therefore, we hypothesized that the inactivation of MEL-11 might enhance the preferential engulfment of lagging univalents by the first polar body. Unlike NMY-2 depletion, which generates 100% dead embryos, the lethality of *mel-11* mutants is rescued by wild-type sperm so the chromosome constitution of progeny from a *mel-11* mother can be scored by phenotype. Mating otherwise wild-type males bearing the recessive X-linked marker, *lon-2*, to *him-8* hermaphrodites allows measurement of the frequency of nulloX ova (which give rise to lon male progeny) and diploX ova (which give rise to XXX dumpy progeny) (*Hodgkin et al., 1979*). Random segregation of univalents should generate a 1:1 ratio of nulloX:diploX ova. We found that *mel-11* increased the segregation bias of *him-8* by sevenfold from 3:1 to 23:1 (*Table 1*). This result indicates that more rapid furrow ingression captures more lagging univalents in the first polar body resulting in more nulloX ova.

**Table 1**. Enhancement of the segregation bias in *him-8* mutants by mutations in the myosin phosphatase, *mel-11*

**Self-progeny counts**

| Genotype | Temperature (°C) | % XO male | % XX hermaphrodite | % XXX Dpy | Total progeny |
|---|---|---|---|---|---|
| *mel-11(sb55) unc-4* | 20 | 0.2 | 99.8 | NC | 1763 |
| *mel-11(sb55) unc-4; him-8* | 20 | 49* | 51 | NC | 374 |
| *unc-4; him-8* | 20 | 34 | 66 | NC | 1442 |
| *mel-11(it126) unc-4* | 15 | 0.6 | 99 | NC | 790 |
| *mel-11(it126) unc-4; him-8* | 15 | 58* | 38.6 | 3.4 | 873 |

**Ratio of nulloX ova/diploX ova calculated from progeny of cross with *lon-2* males**

| Maternal genotype | Temperature (°C) | # NulloX (ion male progeny) | # DiploX (dpy progeny) | Nullo/diplo | Total progeny |
|---|---|---|---|---|---|
| *mel-11(it26) unc-4* | 25 | 1 | 0 | NA | 785 |
| *mel-11(it26) unc-4; him-8* | 25 | 160 | 7 | 22.9 | 595 |
| *unc-4; him-8* | 25 | 98 | 31 | 3.2 | 677 |

*mel-11* increases the frequency of male progeny from *him-8* mothers. *mel-11(sb55)* and *mel-11(it26)* worms produce high frequencies of dead embryos, which cannot be scored for sex at 25°C (*Wissmann et al., 1999*). Percent male (XO), hermaphrodite (XX), and dumpy (XXX) progeny from self-fertilizing *mel-11*, *him-8*, or *him-8 mel-11* double mutant worms were therefore scored at 15°C and 20°C. Only progeny that developed to the L4 or adult stage were counted. *Two-tailed p < 0.0001 by binomial test compared with *him-8* alone. 100% of *mel-11(it26)* self progeny die as embryos at 25°C, but this lethality is rescued by *mel-11(+)* sperm (*Kemphues et al., 1988*). The progeny of *mel-11(it26)* hermaphrodites crossed with *lon-2* males could therefore be scored at 25°C. When *lon-2(+)* hermaphrodites are crossed with *lon-2* males (*lon-2* is a recessive X-linked marker), 50% of the ova will be fertilized by sperm with a single *lon-2* X chromosome. Fertilization of a nulloX ova by a *lon-2* X sperm will result in a *lon-2* male. Fertilization of a diploX ova by a *lon-2* X sperm will result in a XXX dumpy worm. Random segregation of the unpaired X chromosomes in *him-8* would result in a ratio of nulloX/diplo X ova of 1.0. The *mel-11; him-8* double mutant showed a sevenfold increase in the ratio of nullo/diploX ova relative to *him-8* alone, indicating an increased efficiency of eliminating maternal unpaired X chromosomes.

To test our hypothesis that a lagging chromosome inhibits contractile ring scission to allow univalent capture, we asked whether the presence of late-lagging univalents might cause misplacement of the contractile ring from the 50% spindle length scission point observed in wild-type embryos (*Fabritius et al., 2011b*) by time-lapse imaging of the plasma membrane marker GFP:: PH (*Figure 5A*). Spindle length was measured between the outside edges of the main chromosome masses and only in frames in which both chromosome masses were in focus (*Figure 5C*). In *him-8* embryos, when there were no lagging univalents, the contractile ring ingressed normally to 50% spindle length as measured from the outside edge of the main chromatin mass in the interior of the embryo. When a lagging univalent was seen segregating into the polar body, the contractile ring was seen ingressing deeper into the embryo to 40% spindle length (*Figure 5C,D*). Alternatively, when a lagging univalent was seen segregating into the embryo, the contractile ring ingressed inward to a shallower depth at 59% spindle length (*Figure 5C,D*).

To further test the idea that a late-lagging univalent might influence the choice of the scission point, we imaged formation of the first polar body in wild-type or *him-8* worms expressing GFP:UNC-59 (septin) and mCherry: histone. Septins are polymerizing GTPases that assemble in the contractile ring with myosin II, F-actin, and anillins (*Green et al., 2013*). In wild type or *him-8* with early-resolving univalents, GFP:UNC-59 labeled a flat washer–shaped contractile ring that moved down to the midpoint of the elongating anaphase spindle as reported previously for GFP:NMY-2 (*Fabritius et al., 2011b*). The septin ring transformed into a tube (*Figure 6A*, 255 s) as previously described for myosin and ANI-1 (*Dorn et al., 2010*). When cortical furrowing relaxed at the end of telophase I, the septin tube moved outward to the embryo surface, then flopped over, and remained as a separate entity next to the chromosomes in the first polar body (*Figure 6A*, 420 s). In 5/10 *him-8* embryos in which the septin-labeled ring reached the lagging univalent, the univalent was trapped in the septin tube and moved with the septin tube outward during cortical relaxation (*Figure 6B*). In these cases, the univalent remained trapped in the septin tube adjacent to the polar body as shown in *Figure 6B,C*. In 3/10 cases, the septin ring passed the univalent before the tube was formed, and in these cases, the univalent joined the main mass of chromatin in the polar body. In the 2/10 cases where the univalent did not end up in the polar body, the univalent slipped out of the septin tube into the embryo before the septin tube moved toward the embryo surface. These results are consistent with a model where the septin tube traps late-lagging univalents until scission occurs on the embryo side of the septin tube.

To further test the idea that late-lagging univalents are physically trapped in the septin tube, we tried to influence the integrity of the septin tube without blocking polar body scission. Septins act together with anillins (*Green et al., 2013*). *C. elegans* has three anillins: ANI-1 that is required for polar body scission, ANI-2 that is required for gonad development, and ANI-3 that has no known function (*Maddox et al., 2005*). We hypothesized that ANI-3 might play a non-essential structural role in the polar body septin tube and that *ani-3*(*RNAi*) might therefore allow late-lagging univalents to slip out of the tube back into the embryo. Indeed, RNAi of ANI-3 initiated on L4 *him-8* hermaphrodites (which have already completed spermatogenesis) significantly (p < 0.001 binomial test) reduced the fraction of XO male progeny from 37% (n = 9 mothers, 1960 progeny) to 27% (n = 11 mothers, 2123 progeny), whereas *ani-3* (*RNAi*) had no significant effect on wild-type worms (wt: 0.04% XO, 1% dead, n = 11 mothers; *ani-3* [*RNAi*]: 0.05% XO, 1% dead, n = 17 mothers). This result suggests that compromising the integrity of the septin tube may reduce the efficiency of trapping lagging univalents in the septin tube. ANI-3 depletion did not significantly increase the frequency of XXX dumpy progeny from *him-8* mothers (*him-8*: 3% XXX, 5% dead; *him-8 ani-3*[*RNAi*]: 4% XXX, 6% dead). This apparent inconsistency might be explained if additional ANI-3–independent mechanisms act to reduce the number of XXX progeny (see below).

## The early anaphase segregation bias

Lagging chromosomes were resolved prior to contractile ring ingression in 45% of embryos with lagging chromosomes at anaphase I. These were resolved toward the cortex 64% of the time (n = 72), and NMY-2 depletion had no significant effect on this class of embryos (p = 0.8) (*Figure 5B*). These results suggest that an additional mechanism biasing univalent movement toward the cortex might be at work earlier in the cell cycle. During wild-type meiosis, bivalents congress to the metaphase plate on an 8-µm long spindle that is oriented parallel to the cortex. Upon anaphase promoting complex activation, the meiosis I spindle shortens to 4.8 µm in length (*Yang et al., 2003*), then one spindle pole

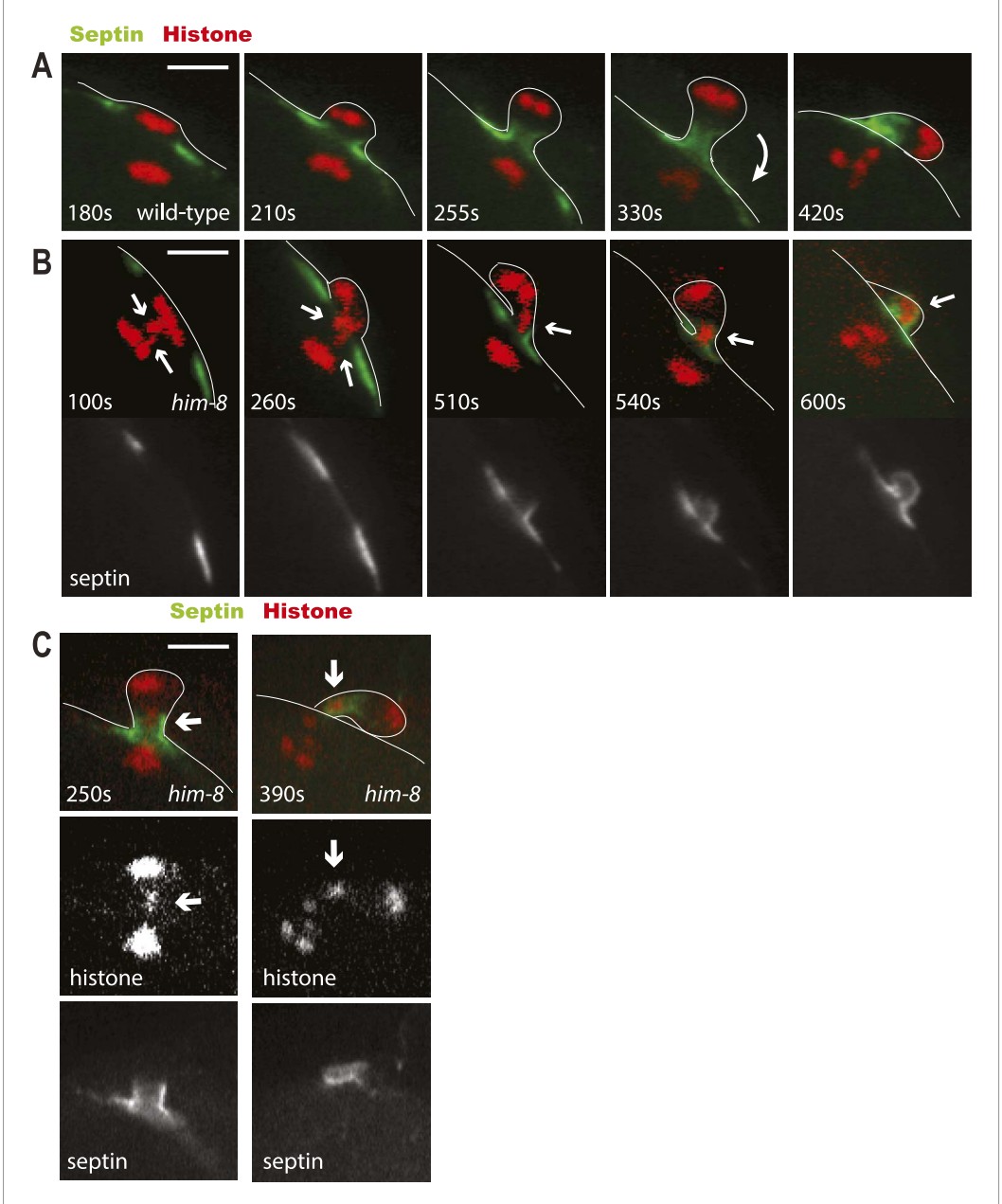

**Figure 6**. Lagging chromosomes are captured by the septin tube and expelled with polar bodies. Time-lapse imaging of embryos expressing GFP::septin and mCherry::histone. (**A**) Time-lapse images of a living wild-type embryo undergoing anaphase I show the conversion of a flat washer–shaped contractile ring into a tube during formation of the first polar body. (**B**) Time-lapse images of a living *him-8* embryo show two lagging chromosomes at anaphase I (arrows) as one moves into the polar body early on, while the second is trapped in the septin tube and is extruded with the first polar body. (**C**) 2 time points of a *him-8* embryo during telophase I showing the lagging chromosome trapped in the septin tube. Bar = 4 μm. Times are from the onset of homolog separation.

moves to the cortex in a dynein-dependent manner, and homolog separation initiates (*Ellefson and McNally, 2009*, *2011*). We found that univalents were misaligned toward the spindle poles in fixed *him-8* embryos at late metaphase I, when the meiotic spindle is shortened but not yet rotated (*Figure 7A–B*). In 46% of these embryos, both univalents were misaligned toward the same pole (*Figure 7B*), close to the 50% expected from random positioning. In fixed *him-8* embryos at the onset of anaphase, when spindles are shortened and rotated but chromosomes are not yet separated, 57%

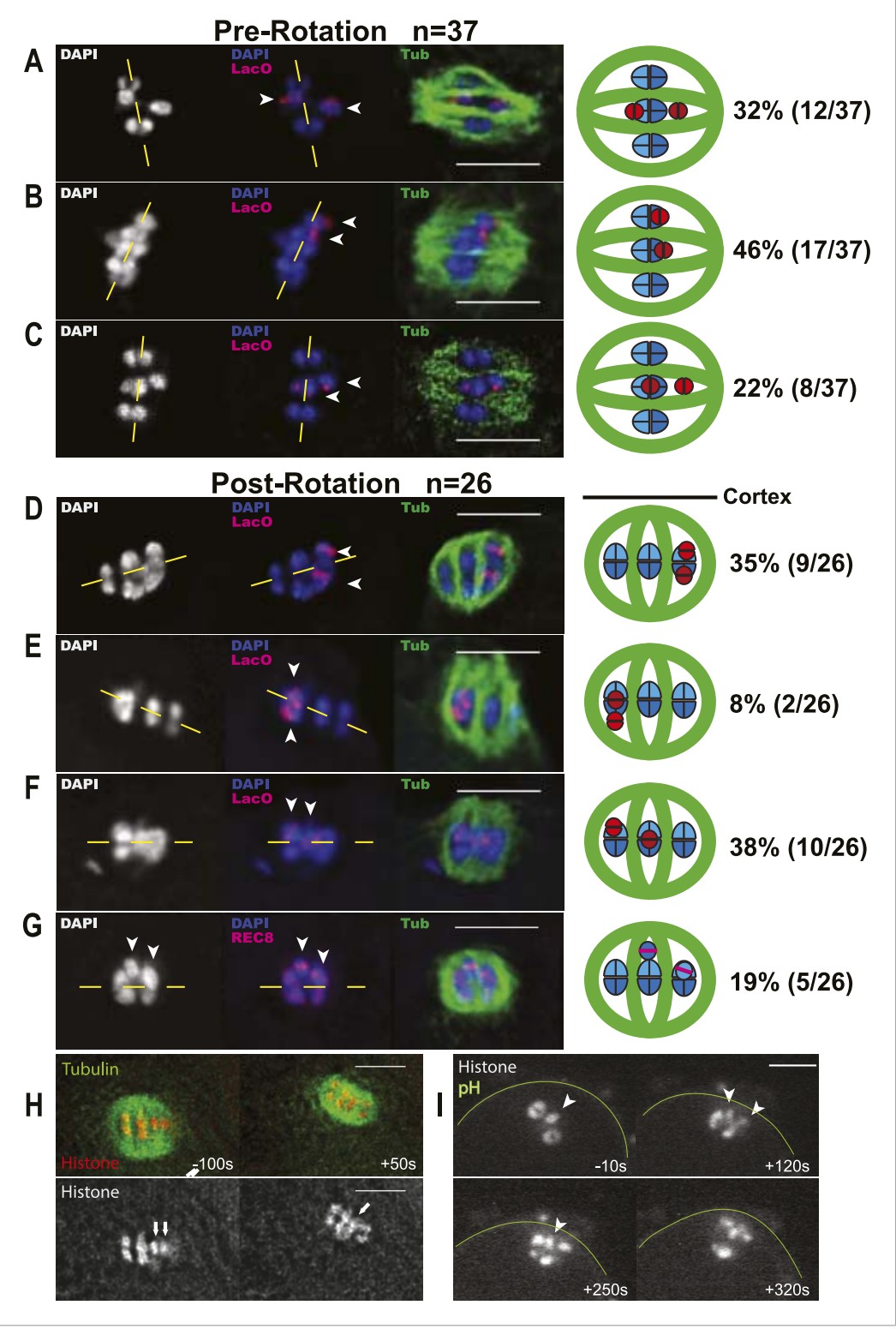

**Figure 7**. Early bias of univalent X chromosomes might occur at the metaphase to anaphase I transition. Representative cartoon diagrams and Z projections from fixed embryos stained with DAPI, anti-tubulin antibody, and LacO(X) FISH probe. Cortex is at the top. (**A–C**) Both X univalents on metaphase I spindles that were shortened (5.3–7.2 μm spindle length) but still parallel to the embryo cortex were frequently (46%) aligned closer to the same spindle pole. (**D–G**) One or both univalents on MI spindles that had rotated but homologs had not yet separated

*Figure 7. continued on next page*

*Figure 7. Continued*

were closer to the cortex and future polar body in 38 + 19% of embryos. Both univalents were never observed closer to the interior spindle pole. Yellow dashed lines indicate the metaphase plate. (**H** and **I**) Time-lapse images of two univalents (arrows in **H**) or one univalent (arrowhead in **I**) offset from the metaphase plate just before rotation of the univalent-proximal pole to the cortex. Time zero is initiation of spindle rotation. Bar = 5 μm.

had one or both univalents closer to the cortical pole (38 + 19%; *Figure 7F,G*). No embryos had both univalents closer to the interior spindle pole. We hypothesized that one of two mechanisms might link spindle rotation with the early anaphase preference for univalent movement toward the cortex. Univalents might stochastically align closer to one spindle pole before rotation and bias the movement of that pole to the cortex. Alternatively, the cortex-proximal pole might acquire distinct biochemical properties after rotation due to cortical contact and subsequently generate more pulling force on the lagging univalents and pull them preferentially toward the cortex.

To test whether spindle rotation is involved with the *him-8* segregation bias, we utilized *mei-2* (*ct98*), a partial loss of function katanin mutant, which causes a failure of meiotic spindle rotation but still allows polar body formation and production of viable progeny (*McNally et al., 2006*). If offset univalents bias spindle rotation or if the cortex-proximal pole exerts greater pulling on univalents after rotation, then a *mei-2*(*ct98*) *him-8* double mutant should have a reduced frequency of male progeny relative to *him-8* alone. At 20°C, the *mei-2*(*ct98*) *him-8* double mutant produced only 21% male progeny (n = 1440 progeny from 14 parents), which is significantly less than the 36% male progeny produced by the *him-8* single mutant (n = 964 progeny from 8 parents; p < 0.0001 by one-tailed binomial test) and is significantly different than *mei-2* (*ct98*) alone (0% males; n = 925 progeny from 8 parents). The reduction in male progeny is unlikely to be due to effects on spermatogenesis, as sperm is unaffected by katanin-null mutants (*Mains et al., 1990*). This result is consistent with either spindle rotation-based models for the early anaphase segregation bias. The role of spindle rotation is not conclusive; however, since *mei-2*(*ct98*) meiotic spindles have other phenotypes besides spindle rotation failure (*McNally et al., 2006*).

Absolute distinction between the two possible rotation models would require unambiguous tracking of both univalents before, during, and after spindle rotation. This was not possible in any of 201 time-lapse sequences. In 8 particularly clear time-lapse sequences, one or both univalents could be identified unambiguously 10–100 s before initiation of spindle rotation. In 6/8 of these cases, the univalent-proximal pole rotated to the cortex (*Figure 7I,H*). In 1/8 cases, the univalent-proximal pole rotated away from the cortex. In 1/8 cases, the two univalents were offset to opposite poles both before and during rotation. If the cortical environment conferred a stronger pulling force on the cortical pole after rotation, then lagging univalents crossing the midpoint of the anaphase spindle should be common. Time-lapse imaging of spindles after rotation revealed that among 30 embryos in which one or two lagging chromosomes were already positioned closer to one pole at anaphase I onset and the lagging chromosome resolved early, the lagging chromosome resolved toward the pole that it was already close to in 80% of these embryos (data not shown). Among the 20% of embryos in which the lagging chromosome moved to the opposite pole after spindle rotation, the chromosome moved toward the cortex 3 times, toward the embryo 3 times, and in one instance, the two lagging chromosomes resolved to opposite poles. These observations are not consistent with a cortical pole that generates a stronger pulling force after rotation but instead favor the idea that the offset position of univalents before rotation increases the probability that the univalent-proximal pole will move to the cortex.

Two results suggested that additional factors might contribute to the overall inheritance of univalent X chromosomes. Both *ani-3*(*RNAi*) and *mei-2*(*ct98*) reduced the frequency of male self-progeny from *him-8* worms without increasing the frequency of triploX self-progeny. We therefore tested whether aneuploid sperm might influence phenotypic outcomes by LacO(X) FISH on pronuclear stage embryos from self-fertilized *him-8* mothers (not shown). Before pronuclear meeting, male pronuclei are distinguished from female pronuclei by the presence of sperm asters. We observed 90% haploX, 8% nulloX, and 2% diploX male pronuclei (n = 52). These values are significantly different than the 50%, 25%, 25% expected from random segregation (two-tailed p < 0.0001 by chi square) and are similar to the frequencies obtained by *Hodgkin et al. (1979)* using genetic tests with sex-reversed

*him-8* XX males. 10% nulloX sperm thus make a small contribution to reducing the frequency of XXX self-progeny.

## Discussion

*Hodgkin et al. (1979)* showed that *C. elegans* that are trisomic for the X chromosome or that fail to form a chiasma between the normal two X homologs have fewer trisomic offspring than expected from random segregation. Our results explain this phenomenon by demonstrating that two cellular pathways preferentially segregate X univalents into the first polar body. Mechanisms reducing the frequency of trisomic offspring have not been investigated in other model organisms because in both mouse and *Drosophila*, animals with trisomy X are not fertile (*Schupbach et al., 1978*; *Tada et al., 1993*), and there are no mutants, like *him-8*, that specifically block crossover formation on one specific chromosome in females. However, women with trisomy 21 or trisomy X are often fertile and have been reported to have more than 50% euploid offspring (*Bovicelli et al., 1982*; *Neri, 1984*; *Ratcliffe et al., 1991*; *Robinson et al., 1991*; *Stewart et al., 1991*). Triploid oysters provide a stronger example of apparent female-specific correction to a diploid state. Eggs produced by triploid females and fertilized with sperm from diploid males result in 57% diploid, 31% triploid, and 12% aneuploid offspring, whereas eggs produced by diploids and fertilized by sperm from triploids result in 15% diploid and 85% aneuploid offspring (*Gong et al., 2004*). Gaging the likelihood that the phenomenon described here for *C. elegans* might be relevant to trisomic humans or triploid oysters is complicated by two issues. In contrast with trisomic *C. elegans*, triploid oysters (*Guo and Allen, 1994*) and trisomic human oocytes sometimes form trivalent structures rather than a separate bivalent and univalent. Only 42–16% of diplotene oocytes from fetuses with trisomy 21, trisomy 13, or trisomy 18 exhibited a separate bivalent and univalent (*Roig et al., 2005*; *Robles et al., 2007*). It is difficult to predict the behavior of trivalents on the spindle. In addition, it is not clear whether a univalent present during anaphase I of a human or oyster oocyte would be more likely to move to one pole intact as in *C. elegans* or to separate equationally. We speculate that single chromatids resulting from equational separation of univalents at anaphase I could be subjected to asymmetric segregation at anaphase II. Our results suggest that any chromosome that lags during late anaphase might be preferntially expelled simply due to the conserved asymmetric nature of polar body formation.

There is one example where a univalent chromosome exhibits the opposite of the segregation bias reported here in *C. elegans*. In the 44–78% of oocytes from XO mice in which the univalent segregates intact at anaphase I, the univalent is preferentially retained in the embryo (*Lemaire-Adkins and Hunt, 2000*). This appears to be a difference between worms and mice rather than a difference between a trisomy and a monosomy since sex-reversed XO *C. elegans* produce an excess of nulloX ova (*Hodgkin, 1980*).

Discerning the overall significance of preferentially placing univalents into the first polar body is a complex problem. In the case of an XXX mother or a mother with a mosaic ovary containing trisomic and diploid oocytes, these pathways would increase the frequency of normal haploid eggs relative to that expected from random distribution of a single univalent (*Figure 1*). In the case of diploid oocytes with two univalent autosomes, however, 100% efficient expulsion of univalents into the first polar body would increase the frequency of lethal monosomy. Data shown in *Figure 2*, however, show no significant decrease in haploid eggs (interpreted from the frequency of MII spindles with 6 chromosomes) from *him-8* or *zim-2* mothers relative to the 50% that would occur by random distribution. Thus, the efficiency of placing univalents in the first polar body has evolved to a point that corrects trisomy without reducing the frequency of haploid eggs produced from oocytes that failed to form a chiasma between one pair of homologs. The conservation of these mechanisms in other species will have to be elucidated by studies focused specifically on the concept of chromosomal errors that are corrected, rather than caused, by female meiotic spindles.

## Materials and methods

### Worm strains

The genotypes of *C. elegans* strains used in this work are listed in *Supplementary file 1*. For LacO(X) FISH, EG7477, which has lac operator arrays integrated on chromosome II and X, was outcrossed to *him-8* males or to wild-type males to eliminate the extra LacO array on chromosome II, generating strains FM299 (wild-type LacO(X)) and FM300 (*him-8* lacO(X)). The loss of the chromosome II LacO array and homozygosity for the X chromosome array were confirmed by PCR.

## RNAi

RNAi was carried out by feeding bacteria (HT115) induced to express double-stranded RNA (*Timmons et al., 2001*). The clones used were *nmy-2* I-3L24, *ani-3* V-12J23 (*Kamath et al., 2001*).

## Live imaging

Adult hermaphrodites were anesthetized with tricaine and tetramisole and immobilized between a coverslip and agarose pad on a slide. The time-lapse images shown in *Figure 2A–E* and *Figure 4A–B* were captured on an Olympus (Center Valley, PA) IX71 microscope equipped with a 60× PlanApo NA 1.42 oil objective and an ORCA R2 CCD camera (Hamamatsu Photonics, Hamamatsu City, Japan). Hg arc excitation light was shuttered by a Sutter Lambda 10-3 shutter controller (Sutter Instruments, Novato, CA). Images shown in *Figure 5A* were captured with an Intelligent Imaging Innovations (Denver, CO) Marianas Spinning Disk Confocal equipped with a Photometrics (Tucson, AZ) Cascade QuantEM 512SC EMCCD, and Zeiss 63× 1.4 objective. Image sequences in *Figure 6* were captured with a Perkin Elmer-Cetus (Waltham, MA) Ultraview Spinning Disk Confocal equipped with an Orca R2 CCD and an Olympus 60× 1.4 objective.

## Immunofluorescence

Meiotic embryos were extruded from hermaphrodites by gentle squishing between coverslip and slide, flash frozen in liquid $N_2$, permeabilized by removing the coverslip, and then fixed in cold methanol before staining with antibodies and DAPI. Antibodies used in this work were mouse monoclonal anti-tubulin (DM1alpha, 1:200; Sigma), mouse monoclonal DM1alpha:FITC conjugated (1:30; Sigma), rabbit anti-REC-8 (from 1:500; Josef Loidl), Alexa 594 anti-rabbit, and Alexa 594 anti-mouse (both from Molecular Probes and used at 1:200). Images in *Figure 3* were captured with an Applied Precision Deltavision Deconvolution system equipped with an Olympus PlanApo 60× 1.40 objective and a CoolSnap HQ CCD camera (Photometrics). Deltavision z-stacks were captured at 130-nm intervals. Images in *Figures 1, 2F–J, 4C–E, 7* and *Figure 2—figure supplement 1* were captured with the Olympus IX71 described above but using an Olympus DSU (disc scanning unit). Z stacks were acquired by taking images every 200 nm (unless otherwise noted) from the top to the bottom of the spindle tubulin signal.

## Deconvolution

Deconvolution was performed on most images shown. Deconvolution of time-lapse movies from the IX71 was performed using Huygens Professional X11 (Scientific Volume Imaging, Hilversum, Netherlands), with point spread functions determined from bead images. Deltavision z-stacks were deconvolved using Softworx native deconvolution software, with PSFs calculated from bead images taken on that system.

## IF-C-FISH: immunofluorescence with chromosome FISH

A lac operator oligonucleotide CCACATGTGGAATTGTG AGCGGATAACAATTTGTGG and an oligonucleotide corresponding to an X-specific repeat, XC (*Phillips et al., 2005*) TTTCGCTTAGAGC GATTCCTTACCCTTAAATGGGCGCCGG, were each synthesized with 3′ and 5′ Texas Red and used in hybridization to LacO arrays integrated on X or V or to endogenous X sequences. FISH with immunofluorescence was performed as described by *Phillips et al. (2009)* with some modifications.

Worms were washed in 0.8% egg buffer and then placed on slides pre-coated with poly-L-lysine (Sigma). Worms were then gently crushed between the slide and a 25-mm sq. #1 coverslip to extrude meiotic embryos and immediately submerged in liquid nitrogen for 10–15 min. Coverslips were then flicked off to freeze-crack eggshells, and slides were submerged in −20°C methanol for 20–30 min. Slides were then washed in 1× phosphate buffered saline (PBS) twice for 10 min and then in 1× PBST (PBS with 0.2% Tween-20) for 10 min. Slides were then blocked in 1× PBST with 4% bovine serum albumen (BSA) for 30–45 min at room temperature in a moist chamber. Blocking solution was wicked away being careful not to dry out the samples, and FITC-conjugated DM1a was applied 1:30 in 1× PBST with 4% BSA using 20 μl coverwells (Grace BioLabs). Slides were incubated in this antibody for 4 hr at room temperature or left overnight at 4°C. Slides were then washed sequentially in 1× PBST, 1× PBS, and 2× SSCT (saline sodium citrate buffer with 0.4% Tween-20) for 10 min each. Following the last wash, slides underwent secondary fixation in 7% formaldehyde in 1× egg buffer for 5 min and were immediately dipped in 2× SSCT to wash off fixative. Slides were then washed in 2× SSCT twice for 5 min

each and then pre-hybridized. Pre-hybridization was performed by adding 200 µl of 50% formamide in 2× SSCT with a 200 µl coverwell (Grace BioLabs) overnight at 37°C in a moist chamber. After 24 hr, slides were taken out of 37°C incubation and placed at room temperature while the FISH probe was prepared. The FISH probe was prepared by adding 0.6 µl of the stock (900 ng/µl) to 30 µl of hybridization buffer (hybridization buffer was made as described in *Phillips et al., 2009*) with 300 µl/ml salmon sperm DNA and 0.1% Tween-20 per slide. Slides were then incubated in 30 µl of this solution under a hybridization slip (Grace BioLabs) at 95°C for 3 min on an OmniSlide (Thermo Scientific) and then at 37°C in a moist chamber for 48–72 hr. Following this incubation, slides were washed in 50% formamide in 2× SSCT as before but for two 1-hr incubations. Finally, slides were stained with DAPI by submerging in a Coplin jar filled with 2× SSCT 6 µg/ml DAPI for 10 min and were then washed for 30 min in fresh 2× SSCT. Slides were then wicked dry with a Kimwipe taking care not to dry out the sample and were mounted with 8 µl of DABCO Mowiol and sealed with nail polish. Following 2–3 days for curing, slides were imaged.

### Metaphase chromosome counts

Chromosome counts were carried out on live embryos in utero or on fixed embryos extruded from the worm by locating metaphase spindles whose chromosomes were all aligned at the metaphase plate. Z stacks were captured at 200-nm intervals. Spindles that were oriented sideways, with their metaphase plates perpendicular to the imaging plane, were reconstructed using ImageJ 3D stack reconstruction, and chromosomes were counted only if individual masses could be discerned. Metaphase II spindles were distinguished from metaphase I spindles by the presence of polar bodies.

### Analysis of lagging chromosome resolution

Time-lapse images of lagging chromosomes in FM125, FM126, and FM232 were acquired at 10-s intervals beginning at late metaphase I when the spindle is shortening and rotating and continuing through polar body extrusion and the formation of the metaphase II spindle. The direction of resolution was determined from the last frame where the lagging chromosome was still discernable from the segregating chromosome masses. At this frame, spindle length was determined by measuring the distance between the outside edges of the main masses of segregating chromosomes. Lagging chromosomes that decided which way to go when the spindle was more than 4 µm long were classified as late resolving because earlier work indicated that myosin-dependent polar body scission occurs when spindles are longer than 4 µm (*Fabritius et al., 2011b*). We confirmed this assumption by finding that 5/5 FM232 (GFP:PH) spindles longer than 4 µm exhibited deep cortical furrows.

For *nmy-2(RNAi)* time-lapse sequences, only embryos in which polar body extrusion completely failed were analyzed. The fate of lagging chromosomes was scored based on whether they ended up at the cortex or in the interior prior to the formation of the metaphase II spindle. Often, chromosomes at the cortex were picked up by the metaphase II spindle. These were still scored as cortex-fated lagging chromosomes.

## Acknowledgments

We thank the following for *Caenorhabditis elegans* strains: Christian Frøkjær-Jensen, Erik Jorgensen, Aaron Severson, Barbara Meyer, Anne Villeneuve, Amy Maddox, Becky Green, Karen Oegema, Paula Checci, and the *Caenorhabditis* Genetics Center, which is funded by the NIH Office of Research Infrastructure Programs (P40 OD010440). This work was supported by National Institute of General Medical Sciences Grant 1R01GM-079421 (to FJM), a grant from the Canadian Institutes of Health Research (To PEM), a fellowship from the Floyd and Mary Schwall (to DC), and NIH training grant T32 GM007377 (to DC). We also thank Dan Starr, JoAnne Engebrecht, Amy Fabritius, Lesilee Rose, and Jon Flynn for critical reading of the manuscript.

## Additional information

### Funding

| Funder | Grant reference | Author |
| --- | --- | --- |
| National Institute of General Medical Sciences (NIGMS) | 1R01GM-079421 | Francis J McNally |
| Canadian Institutes of Health Research | Title: Genetic analysis of the C. elegans cytoskeleton | Paul E Mains |

| Funder | Grant reference | Author |
|---|---|---|
| National Institutes of Health (NIH) | Training Grant: T32 GM007377 | Daniel B Cortes |
| University of California, Davis | Floyd and Mary Schwall fellowship | Daniel B Cortes |
| National Institutes of Health (NIH) | Office of Research Infrastructure Programs P40 OD010440 | Francis J McNally |

The funders had no role in study design, data collection and interpretation, or the decision to submit the work for publication.

## Author contributions

DBC, KLMN, PEM, Conception and design, Acquisition of data, Analysis and interpretation of data, Drafting or revising the article; FJMN, Conception and design, Analysis and interpretation of data, Drafting or revising the article

## Additional files

### Supplementary files

• Supplementary file 1. *C. elegans* strains used in this study.

• Supplementary file 2. Z-stack of XC FISH on XXX wild-type metaphase plate in meiosis I. 16-bit 3-channel TIFF can be opened using FIJI or basic ImageJ (http://fiji.sc/Downloads). Data shown are a z-stack acquired with 300 nm steps through a meiosis I metaphase spindle. Chromosomes are shown in blue (DAPI), tubulin antibodies label the spindle in green, and the XC FISH probe labels X chromosomes (2 present) in red. Channels can be split for individual analysis using the channel splitter (Image > Colors > Split channels) or can be hidden using the channels tool (Image > Colors > Channels tool).

• Supplementary file 3. Z-stack of XC FISH on XXX wild-type metaphase plate in meiosis II. 16-bit 3-channel TIFF can be opened using FIJI or basic ImageJ (http://fiji.sc/Downloads). Data shown are a z-stack acquired with 300 nm steps through a meiosis II metaphase spindle. Chromosomes and the first polar body, which is on the top, are shown in blue (DAPI), tubulin antibodies label the spindle in green, and the XC FISH probe labels X chromosomes (1 present on the spindle) in red. Channels can be split for individual analysis using the channel splitter (Image > Colors > Split channels) or can be hidden using the channels tool (Image > Colors > Channels tool).

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
