## [Decision Letter]

Thank you for sending your work entitled “The asymmetry of female meiosis reduces the frequency of inheritance of unpaired chromosomes” for consideration at *eLife*. Your article has been favorably evaluated by Stylianos Antonarakis (Senior editor) and two reviewers, as well as a member of our Board of Reviewing Editors.

Both reviewers were generally positive about your paper, but one of them had some concerns about your interpretation.

Your data clearly show that there is increased probability for unpaired chromosomes to be segregated into the polar body compared to the zygote. The FISH analysis leaves no doubt that the unpaired chromosome is the one being asymmetrically distributed during meiosis I. However, the major contributing factor to the asymmetry appears to be the inability of unpaired chromosomes to lose cohesion during meiosis I. This aspect needs to be paid greater attention as it is arguably a prerequisite for any type of asymmetric distribution to occur. Given recent work from the Meyer lab, the most likely explanation for this is that in the absence of recombination the cohesion is largely comprised of complexes containing Rec8 (deposited during pre-meiotic S-phase), which are more difficult to cleave in meiosis I (on bivalents, the axis of cohesion loss during meiosis I that is defined by crossovers has primarily Coh3/4-containing cohesion; Rec8 and Coh3/4 are different kleisin subunits of the cohesion complex that are employed during meiosis). Another explanation is that the lack of chiasma somehow reduces Aurora B loading onto the chromosome axis and prior work has shown that Aurora B is important for segregation. A supplementary figure shows reduced Aurora B on univalents in meiosis I compared to bivalents and also compared to paired sister chromatids in meiosis II. As mutants in Rec8 show equational division and form viable triploids (when mated to a wild-type male), I suspect the former may be the reason that the unpaired X chromosomes do not show equational meiosis I. Thus, it is not the “asymmetry of female meiosis” but the “inability to dissolve meiotic cohesion and the asymmetry of female meiosis” that is at play here. To address this point, we suggest the following experiment:

An experimental test would be to make univalents by Rec8 inhibition in the *him-8* mutant and confirm by FISH that there is no asymmetry (anticipated from the genetics). One additional issue here is that sperm meiotic defects are not accounted for in the phenotypic outcomes. It would help if the author used FISH to analyze X chromosomes in sperm nuclei and quantitatively relate the combination of their oocyte and sperm analysis to the phenotypic outcome. There may also be some clever genetic way to fertilize with a male that only delivers an X chromosome; this would be even better.

You also examine the potential sources of asymmetry. In these cases, we are unsure precisely how spindle length is measured and whether tilting in the Z-axis is accounted for in all of their measurements. The representative images make it unclear precisely how positions (e.g. of the lagging chromosome) were defined. Could you sort this out?

---

## [Author Response]

*An experimental test would be to make univalents by Rec8 inhibition in the* him8 *mutant and confirm by FISH that there is no asymmetry (anticipated from the genetics)*.

[43] showed that the partial depletion of REC-8 from the short arms of a bivalent requires AIR-2. Because removal of the AIR-2 inhibitors, HTP-1/2, from the short arms requires a crossover (Martinez-Perez et al., 2008), it is expected that *him-8* univalents might have both reduced amounts of AIR-2 and increased amounts of REC-8 relative to the short arms of a bivalent. These are not two mutually exclusive possibilities but rather integrally related consequences of a crossover failure. We have added analysis of the fluorescence intensity of REC-8 staining on *him-8* univalents relative to the short-arms of bivalents within the same spindle, along with a citation of [43] who showed that this would be expected when there is reduced AIR-2. We have not added FISH analysis of *rec-8 him-8* double mutants because this would provide limited new insight. Because *rec-8* mutants produce mostly 12 univalents, it is a relatively weak hypothesis that *rec-8 him-8* would be different than *rec-8* alone. Figure 2 in Severson and Meyer (2009) clearly shows that there is no bias toward eliminating univalents at anaphase I in *rec-8* mutants. We agree with the reviewers that univalents that biorient and lose cohesion at anaphase I are likely to show no segregation bias at anaphase I. We believe that equational separation of univalents at anaphase I leaves open the possibility of biased (or random) segregation of single chromatids at anaphase II (this was already stated in our Introduction and is now stated in the Discussion). This cannot be studied in *rec-8* because the second polar body is not formed (Severson and Meyer, 2009).

One additional issue here is that sperm meiotic defects are not accounted for in the phenotypic outcomes.

Sperm meiotic defects caused by *him-8* can only influence the “phenotypic outcome” of self progeny. Our analysis of female meiotic spindles cannot be influenced by the sperm karyotype and is related in the text to phenotypic outcomes from crosses with wild-type XO males. For example, the first comparison of our MII chromosome counts with phenotypic outcomes (beginning of the Results section) refers to phenotypic outcomes from crosses with wild-type males.

*It would help if the author used FISH to analyze X chromosomes in sperm nuclei and quantitatively relate the combination of their oocyte and sperm analysis to the phenotypic outcome. There may also be some clever genetic way to fertilize with a male that only delivers an X chromosome; this would be even better*.

We have added analysis of X chromosome number in male pronuclei of *him-8* self progeny to the last section of the results. This section is focused on looking for additional mechanisms, beyond female meiosis, that might contribute to the low frequency of XXX offspring among self progeny. Our FISH analysis of male pronuclei (distinguishable from female pronuclei by the presence of centrosomes) in the self progeny of *him-8* hermaphrodites revealed 90% haploX, 8% nulloX sperm and 2% diplo X sperm (n=52). This agrees with genetic results in [17] using sex reversed *him-8* XX males and demonstrates that nulloX sperm make a small contribution to the low frequency of XXX self progeny.

You also examine the potential sources of asymmetry. In these cases, we are unsure precisely how spindle length is measured and whether tilting in the Z-axis is accounted for in all of their measurements. The representative images make it unclear precisely how positions (e.g. of the lagging chromosome) were defined. Could you sort this out?

We have added the following sentence in the subsection “The meiotic contractile ring captures lagging X univalents in the first polar body”: “Spindle length was measured between the outside edges of the main chromosome masses and only in frames in which both chromosome masses were in focus (Figure 5)”. Tilting in the Z-axis occurs in some frames of any time-lapse sequence and is obvious because one chromosome mass goes out of focus.